SaPt-CNN-LSTM-AR-EA: a hybrid ensemble learning framework for time series-based multivariate DNA sequence prediction

Yan Wu 1 2 3 wuyan@gnnu.edu.cn
Tan Li 4
Meng-Shan Li 4
Sheng Sheng 1 3
Jun Wang 1 3
Fu-an Wu 1 3 fuan_w@just.edu.cn
1 School of Biotechnology, Jiangsu University of Science & Technology , Zhenjiang , China
2 School of Mathematics and Computer Science, Gannan Normal University , Ganzhou, Jiangxi , China
3 Sericultural Research Institute, Chinese Academy of Agricultural Sciences , Zhenjiang, Jiangsu , China
4 College of Physics and Electronic Information, Gannan Normal University , Ganzhou , China
Fischer Daniel
Electronic publication date: 2023 Oct 4
Publication date: 2023
Volume: 11
Electronic Location ID: e16192
Received 2023 Mar 31; Accepted 2023 Sep 6
Copyright: © 2023 Yan et al.
Copyright year: 2023
Copyright holder: Yan et al.
License: This is an open access article distributed under the terms of the Creative Commons Attribution License, which permits unrestricted use, distribution, reproduction and adaptation in any medium and for any purpose provided that it is properly attributed. For attribution, the original author(s), title, publication source (PeerJ) and either DOI or URL of the article must be cited.
License URL: https://creativecommons.org/licenses/by/4.0/

Keywords: Ensemble learning, DNA sequence, Time series, Biological sequence, Data mining

Funding: National Natural Science Foundation of China 51663001, 52063002, 42061067, 61741202 This work was supported by the National Natural Science Foundation of China (No. 51663001, 52063002, 42061067, 61741202). There was no additional external funding received for this study. The funders had no role in study design, data collection and analysis, decision to publish, or preparation of the manuscript.

==============================
Biological sequence data mining is hot spot in bioinformatics. A biological sequence can be regarded as a set of characters. Time series is similar to biological sequences in terms of both representation and mechanism. Therefore, in the article, biological sequences are represented with time series to obtain biological time sequence (BTS). Hybrid ensemble learning framework (SaPt-CNN-LSTM-AR-EA) for BTS is proposed. Single-sequence and multi-sequence models are respectively constructed with self-adaption pre-training one-dimensional convolutional recurrent neural network and autoregressive fractional integrated moving average fused evolutionary algorithm. In DNA sequence experiments with six viruses, SaPt-CNN-LSTM-AR-EA realized the good overall prediction performance and the prediction accuracy and correlation respectively reached 1.7073 and 0.9186. SaPt-CNN-LSTM-AR-EA was compared with other five benchmark models so as to verify its effectiveness and stability. SaPt-CNN-LSTM-AR-EA increased the average accuracy by about 30%. The framework proposed in this article is significant in biology, biomedicine, and computer science, and can be widely applied in sequence splicing, computational biology, bioinformation, and other fields.

Introduction

Portions of this text were previously published as part of a preprint (https://www.authorea.com/doi/full/10.22541/au.166739767.78591208/v1). Biological sequences are mainly classified into three types: DNA (deoxyribonucleic acids), RNA (ribonucleic acids), and protein sequences. In recent years, biological sequence data mining has been widely concerned and mainly focuses on the prediction and functional analysis of coding and non-coding regions of sequences, sequence analysis, sequence visualization, sequence alignment, gene identification, and evolutionary analysis (Ali et al., 2015; Eisenstein, 2021; Li et al., 2022a; Ullah et al., 2022). The main sequence data mining methods mainly include mathematical statistics method, signal processing method, time series method, and machine learning algorithms (Jiang et al., 2023; Liu et al., 2021; Wang et al., 2022). At present, classification, clustering, alignment, similarity, prediction, and graphical representation of biological sequences have been extensively explored (Abranches et al., 2022; Aevermann et al., 2021; Routhier & Mozziconacci, 2022).

DNA sequences can be regarded as the sequence set of four letters (A, C, G, and T). A time series is a sequence set of digital symbols composed in time sequence and similar to biological sequence in terms of data representation (Boltenkov et al., 2020; Lochel & Heider, 2021; Pavithran et al., 2023; Thuillier et al., 2022). In terms of mechanism, a time series is a sequence based on time, whereas biological sequences also contain a time series relationship related to organism evolution. Time series and biological sequences are related to time order, so biological sequences are similar to time series. A biological time sequence (BTS) is the biological series represented by time series method. In the preliminary work, our research group analyzed relevant literatures on BTS. Based on historical biological sequence data, BTS prediction aims to establish a suitable sequence prediction model from the trend, periodicity, and volatility of biological sequences and then the established model can be used to generate unknown data for biological sequences. Many time series prediction models can be used in BTS analysis. Time series models can be classified into three types: single models, hybrid models, and integrated models (He et al., 2022; Li et al., 2021; Mitra & MacLean, 2021; Wen & Yang, 2021). Single prediction models mainly include autoregressive (AR), moving average (MA), autoregressive moving average (ARMA), autoregressive Integrated moving average (ARIMA), SVM, ANN-based model, and machine learning-based model. Hybrid models can obtain more accurate prediction (Bai et al., 2022; Han et al., 2019; James & Tripathi, 2021; Zhang et al., 2021, 2023). For example, various meta-heuristic algorithms are used to optimize the weights and thresholds of ANN, such as differential evolution (DE), simulated annealing (SA), particle swarm optimization (PSO), and genetic algorithm (GA) (Chu et al., 2022; Gugler & Reiher, 2022; Torkey et al., 2021, 2022; Xia et al., 2022; Zhang, Yan & Aasma, 2020). Integrated models have been widely used in sequence prediction. Integrated models have significant advantages and can improve the accuracy of sequence prediction and reduce the variance. Deep learning algorithms are emerging machine learning algorithms, such as recurrent neural network (RNN) (Li et al., 2022b; Savadkoohi, Oladunni & Thompson, 2021; Yang & Song, 2022; Zhou et al., 2023) and long-short term memory artificial neural network (LSTM) (Anzel, Heider & Hattab, 2022; Jian, Wang & Farimani, 2022; Li et al., 2023; Liu et al., 2022).

Two issues in BST research remain to be addressed. Firstly, biological sequence studies focus on single sequences. One of the disadvantages of single-sequence models is that some important features contained in biological sequences are omitted, thus affecting the modelling effect (Angthong et al., 2020; Du, Du & Li, 2023; Kim et al., 2021; Mondal et al., 2022; Nalecz-Charkiewicz & Nowak, 2022; Namasudra et al., 2023; Thorn et al., 2022). Secondly, although various machine learning algorithms have been widely applied in time series prediction, multivariate biological sequence prediction is still a challenge. In this article, parallel multivariate biological sequences were used for modelling. Next, an integrated model was established through multi-channels to fuse the features of multivariate biological sequences. Finally, a multivariate biological sequence ensemble learning model was proposed based on time series method.

The structure of the article is as follows. In ‘Methodology and Modelling’, the methodology is introduced, including the model framework, construction, and evaluation metrics. In ‘Experiments and Results’, the experiment part is described, including the source of experimental data and primary results of the proposed model. In ‘Discussion’, different metrics with other benchmark models are discussed and the ablation study is provided to verify the contribution of the four modules used in the proposed model. In ‘Conclusions and Outlook’, some conclusions are drawn.

Methodology and modelling

Transformation of biological sequences

A set of DNA sequences with the length n, are denoted as Seq=S1S2⋯Si⋯Sn(Si∈(A,C,T,G)). The following three methods are used to transform biological sequences into time series (Chou, Chen & Huang, 2022; Li, Dai & He, 2022)_ENREF_49.

Spectral time sequence

A spectral time sequence can be expressed with Eq. (1):

(1) x(i)={1,Si=a2,Si=g3,Si=c4,Si=t,i=1,2,…,n.

CGR time sequence

Chaos game representation (CGR) is an iterative mapping technique. It maps each element in a sequence to a continuous coordinate space. The four nucleotides of DNA sequence are represented by the four vertices of a square. The coordinates of each base in the sequence are used to determine the position of the next base as follows:

(1) The coordinates of square vertices are assigned to four nucleotides as: A = (1, 1); C = (−1, −1); G = (−1, 1); T = (1, −1).

(2) The center of the square (0,0) is assigned to the starting position.

(3) The first character of DNA sequence is defined as the current character. The pointer moves half the distance from the current nucleotide to the last nucleotide coordinate point so as to determine the next position.

(4) The next character of DNA sequence is assigned to the current character and then Step 3 is implemented until the end of the DNA sequence. The procedure is illustrated in Eq. (2):

(2) (x(i)=CGRi=CGRi−1−CGRi−1−gi2gi={(1,1),Si=a(−1,1),Si=g(1,−1),Si=c(−1,−1),Si=t)

Z time sequence

In the set of DNA sequences, Ai,Ci,Gi,andTi respectively indicate the numbers of A, C, G, and T in the DNA sequence from Base 1 to Base i. Z sequence is transformed into time series as follows:

(3) {x(i)=Xi+Yi+ZiXi=(Ai+Gi)−(Ci+Ti)Yi=(Ai+Ci)−(Gi+Ti)Zi=(Ai+Ti)−(Ci+Gi)

Model framework and construction

BTS xt is defined as:

(4) xt=Merge(funi(xt−T:t−1),fmulti(xt−T:t−1)),

where xt−T indicates the sequence with a length of T before time t; Merge(⋅) fuses single-sequence pattern funi(⋅) with multivariate-sequence pattern fmulti(⋅). In this article, by modelling funi(⋅) and fmulti(⋅), the ensemble learning module Merge(⋅) is constructed and then the hybrid ensemble learning model (SaPt-CNN-LSTM-AR-EA) is obtained.

The model framework is shown in Fig. 1.

Figure 1 Hybrid ensemble learning framework.

SaPt-CNN-LSTM-AR-EA, hybrid ensemble learning framework.

SaPt-CNN-LSTM-AR-EA has two modules: single-sequence module and multi-sequence module. The input data of SaPt-CNN-LSTM-AR-EA are N biological sequences x(1),⋯x(N). In the single-sequence module, ARFIMA method is used to obtain N single-sequence output vectors. The weighted fusion is carried out to obtain the output of single-sequence module OtAR. In the multi-sequence module, N biological sequences are combined together to form the input of multi-sequence vectors, which are input into LSTM for modelling. Finally, the weighted fusion of multivariate output sequences is carried out to obtain the output of multi-sequence module.

In the single-sequence module, the classical statistical method ARFIMA (Bhardwaj, Gadre & Chandrasekhar, 2020) is used to establish the model. The three parameters of ARFIMA model are optimized with the classical particle swarm evolution algorithm. The particle structure is illustrated as:

(5) ypso1=fpso1(pi,di,qi)

In the multi-sequence module, the temporal convolution of one-dimensional CNN is used to represent the dependence between multi-ple biological sequences. The one-dimensional convolution operation is illustrated as:

(6) C(k)=W(k)×Xt−T:t−1,

where C(k) is the convolution result and W(k) is the k-th convolution kernel.

After the execution of the one-dimensional convolution feature extraction, the results are used as the input of the model LSTM (Karim et al., 2019; Singaravel, Suykens & Geyer, 2018) to obtain the output of the multi-sequence pattern OtLSTM. The three parameters of LSTM model are also optimized by particle swarm optimization algorithm (Bi et al., 2023; Dias et al., 2023; Li, Liu & Wu, 2022). The particle structure is illustrated as:

(7) ypso2=fpso2(Ntr,Nhn,Dtr).

Finally, the output of the frame A is obtained by fusing the output of single sequence and multivariate sequence. The procedure is illustrated as:

(8) Ct(k)=WtAR⊗OtAR+WtLSTM⊗OtLSTM.

Evaluation and testing

Accuracy and correlation

Mean absolute percentage error and correlation coefficient are used to evaluate the prediction accuracy and correlation of the model as follows:

(9) MAPE=1N∑i=1N⁡|yi−y¯iyi|×100%

(10) R2=(∑i=1N⁡(yi−yave)(y¯i−y¯ave))2∑i=1N⁡(yi−yave)2∑i=1N⁡(y¯i−y¯ave)2

where N is the total number of samples; y¯i and y¯ave are respectively the predicted value and predicted average value of the model; yi and yave are the experimental value and mean value, respectively.

Accuracy growth rate

In order to better evaluate the accuracy increase of the integrated model in the prediction process, the accuracy growth rate is used to quantify the accuracy increase as:

(11) PMAPE=|MAPEi−MAPEjMAPEi|×100%

where MAPEiandMAPEj are respectively the mean absolute percentage errors of model i and model j.

Diebold-Mariano test

In order to verify the necessity of the model, the Diebold-Mariano (DM) test is adopted. Significant differences between the models are assessed with the calculated DM values of the prediction errors of the two models. The significance level is set as a. Null hypothesis H0 indicates that the error of the integrated model is not significantly different from the comparison model. Valid hypothesis H1 is an alternative to H0. At the confidence level of 90% (i.e., the significance level a is 0.1), DM value should be less than 1.645. The confidence level of 90% corresponds to 1.96. The confidence level of 99% corresponds to 2.58. Otherwise, H0 is refusedand H1 is accepted.

VAR

Variance of residuals (VAR) is a common indicator to evaluate the predictive stability of a model. Therefore, VAR is used to test the stability of the model as follows:

(12) VAR=(Std(yi¯−yi))2,i=1,2,3,⋯,N.

The larger the VAR value is, the more unstable the prediction result is. In other words, the model is more dependent on samples. On the contrary, the lower dependence of the model on samples indicates the more stable prediction result.

Experiments and results

Experimental data

Source of experimental data

The experimental data were DNA sequences of six viruses (Table 1) downloaded from NCBI (https://www.ncbi.nlm.nih.gov).

Table 1 Source of experimental data.

Label	Source	Accession	Length (bp)	
A	Human adenovirus C	NC_001405	35,937	
B	Dubowvirus MR25	NC_010808	44,342	
C	Infectious bronchitis virus	NC_048213	27,464	
D	Phietavirus MR11	NC_010147	43,011	
E	Abalone shriveling syndrome-associated virus	NC_011646	34,952	
F	Clostridium phage phiCD505	NC_028764	49,316	

Biological sequences are transformed into time series so as to obtain the corresponding BTS. According to the length of each BTS, the sequence is divided into several data to form a BTS database, which is used for model training, verification, and testing. The procedure is illustrated in Fig. 2.

Figure 2 Partition diagram of biological sequence.

After sequence division, each BTS consists of several groups of subsequences. To improve the generalization ability of the model, each subsequence is divided into three subsets: training set (70%), validation set (15%), and test set (15%), as shown in Table 2.

Table 2 Experimental data distribution.

Label	Training set	Validation set	Testing set	Data points	
A	420	89	89	598	
B	519	110	110	739	
C	321	68	68	457	
D	502	107	107	716	
E	408	87	87	582	
F	575	123	123	821	

Stationary analysis

Taking the BTS generated with sequences A, C, and E as an example, the BTS corresponding to the first 3,000 bp is shown in Fig. 3.

Figure 3 (A–C) Biological time sequences.

In Fig. 3, the time series curves obtained with sequences A, C, and E are characterized by large fluctuations, unequal amplitudes, and unequal position intervals, indicating that BTS has typical non-stationary characteristics. Other series have similar non-stationary characteristics.

Pre-processing

In order to reduce the interference caused by non-stationary series, we carried out normalization and variance normalization transformations for each BTS. Each BTS produces six new series: normalized spectral time sequence, variance-normalized spectral time sequence, normalized CGR time sequence, variance-normalized CGR time sequence, normalized Z time sequence, and variance-normalized Z time sequence.

Results

The operating environment is Windows 10 64-bit OS (16 GB of memory and Intel (R) Core™ i7-12700F processor). The deep learning framework was constructed with Matlab2020a. The transformed and preprocessed 6 BTSs were used for model testing and prediction (see the architecture of the model and corresponding parameters is in the Supporting Information). Finally, the output results of the model were reversely normalized to make the model more explanatory.

Firstly, SaPt-CNN-LSTM-AR-EA was trained with the training set. Secondly, various parameters of the model were adjusted to minimize the training error in the training process (see Supporting Information). The relationship between the predicted value and the actual value of each data sample in the training set is shown in Fig. 4.

Figure 4 (A–F) Prediction results of the training set.

The closer the predicted data points are to the experimental line, the smaller the prediction error is. As shown in Fig. 4, the predicted data points of the six biological sequences are basically distributed on the experimental line, indicating that the training effect of SaPt-CNN-LSTM-AR-EA was good on each sequence data set and that the model had been fully trained. The validation set was used to verify the reliability of the model. The parameters of the model were fine-tuned in the validation process so as to reduce the output error of the model. The distribution relationship between the predicted value and the actual value of the model in the validation set is shown in Fig. 5.

Figure 5 (A–F) Prediction results of the validation set.

In the validation set, the data points predicted by the model are basically distributed near the straight line, indicating that the predicted value was in good agreement with the experimental value. The predicted value of SaPt-CNN-LSTM-AR-EA was more consistent with the experimental value, indicating that the trained model had the reliable and accurate prediction ability and could be used to predict BTS.

After training and validation, the prediction experiment was performed with the trained model. With the test set, the model was tested against each biological sequence. The test data of each biological sequence were used for model testing. The prediction results and relevant data statistics of the model are shown in Fig. 6.

Figure 6 Prediction results in the test set.

(A) Mean distribution of the error between predicted and experimental values, (B) error statistics of the model in each biological sequence test, and (C) error distribution of the model in each biological sequence test.

The distributions of predicted values and experimental values and the mean distribution of errors are shown in Fig. 6A. The error statistics and error distribution of the model in each biological sequence test are respectively shown in Figs. 6B and 6C.

The predicted values of each sequence were basically consistent with the actual values except the results of some loci. In the error analysis results, most of the error points were distributed around the value of 0 and the number of sequences with large errors accounted for a small proportion. The predicted values were highly consistent with the experimental values in six biological sequences, indicating that the model had good prediction performance. The performance indexes of the model in the training set, validation set, and test set are shown in Table 3.

Table 3 Predictive performance indexes of the model.

Sequence	Training set	Validation set	Test set	
MAPE	R 2	MAPE	R 2	MAPE	R 2	
A	1.7335	0.9248	1.7491	0.9336	1.9925	0.9105	
B	1.6011	0.9275	1.6279	0.9346	1.6354	0.9167	
C	1.1417	0.9304	1.1331	0.9389	1.5128	0.9268	
D	1.5032	0.9289	1.4305	0.9215	1.6423	0.9142	
E	1.6563	0.9238	1.6101	0.9288	1.8153	0.9186	
F	1.6214	0.9311	1.6042	0.9345	1.7201	0.9193	
Average	1.5429	0.9278	1.5258	0.9320	1.7197	0.9177	

The values of performance indexes fully reflect the comprehensive performance of the model in the three data sets. The error in the validation set was small and the correlation was high. The prediction ability of training and validation sets was better than that of the test set. From the perspective of prediction mechanism, model training and validation aims to reduce the output error, so the correlation coefficient of the training and validation sets should be larger than that of the new samples in the test set.

Discussion

Comparison with other benchmark models

In order to verify the performance of the model, several models with better performances in time series prediction were selected as benchmark comparison models in this article. The theories and parameters of all the models are shown in Table 4.

Table 4 Benchmark comparison models.

Model	Model details	References	
BI-ARFIMA	Bayesian Inference for ARFIMA	Durham et al. (2019)	
ARFIMA-LSTM	ARFIMA-LSTM hybrid recurrent network	Bukhari et al. (2020)	
EA-LSTM	Evolutionary attention-based LSTM	Li et al. (2019)	
CTS-LSTM	LSTM network for correlated time series	Wan et al. (2020)	
Conv-LSTM	Convolutional neural network and LSTM	Fu et al. (2022)	

One hundred samples were randomly selected from six BTS data sets to form the test data set of the comparison model. Each benchmark model was used to predict each sample in the test data set. The prediction results of each model on Sequence A are shown in Fig. 7. The results on other sequences are similar to those of Sequence A.

Figure 7 (A–E) Prediction performance of each benchmark model on sequence A.

The correlation between the predicted and experimental values is shown in Figs. 7A and 7B. The prediction results of the SaPt-CNN-LSTM-AR-EA model were closer to the experimental line, indicating that the predicted results were more consistent with experimental results. The statistical distribution of predicted values shown in Figs. 7C–7E indicated that the SaPt-CNN-LSTM-AR-EA model performed better than other models. The errors of each model are shown in Fig. 8.

Figure 8 (A–C) Test errors of each benchmark model.

In the error curve (Fig. 8A), the error bar of the SaPt-CNN-LSTM-AR-EA model was closest to the origin and the prediction error was smaller, indicating that the model had the highest prediction accuracy. According to the error statistics (Fig. 8B), most of the error points of the SaPt-CNN-LSTM-AR-EA model were distributed between 1.4 and 2.0 and the average error was also the smallest, indicating that the accuracy of the model was relatively high. The correlation coefficient and calculation time of each benchmark model are shown in Fig. 9.

Figure 9 (A and B) Correlation coefficient and calculation time of each compared model.

According to Fig. 9A, the correlation curve of SaPt-CNN-LSTM-AR-EA model is at the top of the coordinate and the coordinate value is closest to 1. In addition, the statistics of correlation data points also showed that the predicted values of the model in this article had the most significant correlation with experimental values. The BI-ARFIMA model had the shortest computation time and CONV-LSTM model had the longest computation time (Fig. 9B). The SAPT-CNN-LSTM-AR-EA model also had the acceptable computation time. Table 5 shows the performance statistics of each model. The SaPt-CNN-LSTM-AR-EA model performed better in terms of both prediction accuracy and correlation and its computation time was also acceptable.

Table 5 Statistics of predictive performance of each benchmark model.

Model	MAPE	R2	Time	
BI-ARFIMA	2.5174	0.8288	10.93	
ARFIMA-LSTM	2.6243	0.8232	21.08	
EA-LSTM	2.4877	0.8149	17.82	
CTS-LSTM	2.2968	0.8673	22.48	
Conv-LSTM	2.2108	0.8664	45.62	
SaPt-CNN-LSTM-AR-EA	1.7073	0.9186	19.77	

The SaPt-CNN-LSTM-AR-EA model had obvious advantages over other models in terms of prediction accuracy and correlation due to the following factors. Firstly, the characteristics of multivariate sequences were fully utilized in the model. Secondly, adaptive pre-training mechanism improved the training performance. Thirdly, the advantages of CNN-LSTM in feature extraction had been fully utilized. The SaPt-CNN-LSTM-AR-EA model belonged to a multi-layer cyclic deep learning framework and fused ARFIMA, so its computation time was long. Compared with CTS-LSTM and CONV-LSTM models, SaPt-CNN-LSTM-AR-EA had a slight advantage in computation time because it adopted one-dimensional convolution operation.

Discussion of different evaluation metrics

Accuracy growth rate

In this article, the accuracy growth rate of each benchmark model was calculated in order to verify the prediction accuracy of the SaPt-CNN-LSTM-AR-EA model on the test dataset. The accuracy growth rate statistics of the SaPt-CNN-LSTM-AR-EA compared with other five benchmark models are shown in Fig. 10.

Figure 10 Accuracy growth rate of the SAPT-CNN-LSTM-AR-EA model compared with other benchmark models.

Compared with the five benchmark models (BI-ARFIMA, ARFIMA-LSTM, EA-LSTM, CTS-LSTM, and Conv-LSTM), the SaPt-CNN-LSTM-AR-EA model had different MAPE growth rates in six different biological sequences. The highest accuracy was increased by nearly 50% and the average accuracy was increased by about 30%. The accuracy improvement was obvious.

The output of the SaPt-CNN-LSTM-AR-EA model had obvious advantages in accuracy due to the comprehensive results of various algorithms. In addition, the performance of the model was significantly improved.

Diebold-Mariano test

DM test was performed to test the model. In invalid hypothesis H0, the error difference of the models is not obvious. In other words, there is no significant difference in prediction accuracy. The DM test values of each model are shown in Table 6. At the significance level of 1%, all DM test values were above the upper limit of 2.58, suggesting that H0 should be rejected and that H1 should be accepted. The DM values indicated that the prediction performance was significantly different among the models.

Table 6 DM values of SaPt-CNN-LSTM-AR-EA and benchmark models (at the significance level of 1%).

Sequence	BI-ARFIMA	ARFIMA-LSTM	EA-LSTM	CTS-LSTM	Conv-LSTM	Average	
A	5.2319	5.2951	5.4511	5.1221	4.2437	5.0688	
B	4.4318	5.3555	5.8193	5.2391	4.1951	5.0082	
C	5.9460	5.4438	4.7947	3.5907	5.2476	5.0046	
D	5.1702	4.9207	5.5258	4.8178	5.0280	5.0925	
E	5.5484	4.9639	4.9925	4.8713	4.6340	5.0020	
F	5.3184	5.3652	4.8212	5.7784	3.8258	5.0218	
Average	5.2745	5.2240	5.2341	4.9032	4.5290	5.0330	

DM test was performed to test the necessity of modelling. DM values at the confidence level of 99% showed that the SaPt-CNN-LSTM-AR-EA model was necessary and effective. The data distribution of DM values suggested the performance differences among the models. The performance of the BI-ARFIMA, ARFIMA-LSTM, and EA-LSTM models was basically the same and the performance of CTS-LSTM was the same as that of CONV-LSTM. CONV-LSTM had a slight advantage. The DM value on each sequence showed that the performance of the model was basically the same among all the sequences.

VAR

The proposed model is to improve the efficiency, accuracy, and stability of prediction. We used variance of residuals (VAR) to test the stability of the model. The VAR change curves of each model are shown in Fig. 11. The VAR curve of the SaPt-CNN-LSTM-AR-EA model was close to the abscissa, indicating that the model was more stable. In addition, BI-ARFIMA and ARFEIMA-LSTM had similar stability and CTS-LSTM was more stable than EA-LSTM.

Figure 11 VAR of each model.

The results of VAR stability test indicated that the stability of the SaPt-CNN-LSTM-AR-EA model was basically the same among all the BTS because the fusion of multivariate sequences in the model avoided the defect of single sequence output error and improved the generalization ability. The results also proved the good scalability of the model.

Discussion of ablation study

To validate the contribution of these modules to the framework, we performed ablation study by respectively removing one module from the four modules in four times. Then, four models were obtained: CNN-LSTM-AR-EA, SaPt-LSTM-AR-EA, SaPt-CNN-LSTM-EA, and SaPt-CNN-LSTM-AR. The performance indexes of these models are shown in Table 7.

Table 7 The results of ablation study.

Removed modules	Model	MAPE	R 2	
SaPt	CNN-LSTM-AR-EA	2.4865	0.7591	
CNN	SaPt-LSTM-AR-EA	4.5621	0.6174	
AR	SaPt-CNN-LSTM-EA	3.8697	0.6726	
EA	SaPt-CNN-LSTM-AR	2.3134	0.7969	
This article	SaPt-CNN-LSTM-AR-EA	1.7197	0.9177	

After any module was removed from SaPt, CNN, AR, and EA, the error was significantly larger and the correlation decreased, indicating that each module contributed to the SaPt CNN LSTM AR-EA model. The SaPt-LSTM-AR-EA model without the CNN module had the worst experimental results, indicating that the CNN module had the greatest contribution. The contributions of the other three modules ranked in the following order: AR > SaPt > EA. The contribution rate is defined as the ratio of the contribution of one module to the total contribution (assuming that the total contribution of the four modules is 100%). Figure 12 shows the accuracy contribution and correlation contribution of each module.

Figure 12 (A and B) Contribution rate of each module.

The contributions of the four modules ranked in the following order: CNN > AR > SaPt > EA and all modules have a contribution rate of over 17% (Figs. 12A and 12B). The ablation study indicated that the four modules were critical and indispensable and jointly contributed to the SaPt-CNN-LSTM-AR-EA model.

Conclusions and outlook

This article proposes an integrated prediction model of biological sequence based on time series theory method: SaPt-CNN-LSTM-AR-EA. The prediction results of DNA sequences of six viruses indicated that the model had high prediction accuracy. The performance analysis and test confirmed the better reliability of the model. This study opened up a new field of BTS research and provided a new idea for biological sequence and time series research. The proposed integrated model framework is significant in many fields, such as biology, computer, economics, and medicine and can be widely used in bioinformation, genetic evolution, financial economy, meteorology, hydrology, signal processing, electric power, medicine, and health care.

The algorithm proposed in this article has achieved good experimental results, but its computing time is long. In future work, we will further improve the performance, efficiency, and generalization ability of the model and reduce computation time and space complexity. We will design more optimized algorithms and models based on the parallel computation strategy for the purpose of mining biological sequence data.

Supplemental Information

Supplemental Information 1 The codes, architecture, parameters, dataset, functions, usage and output of the proposed model.

Click here for additional data file.

Supplemental Information 2 Supplemental Tables.

Click here for additional data file.

Supplemental Information 3 The Theory, Modeling, Results and Validation of Hybrid Ensemble Learning Framework for DNA Sequence Prediction.

Click here for additional data file.

We thank TopPaper for its linguistic assistance in the preparation of this manuscript.

Additional Information and Declarations

Competing Interests

Author Contributions

Data Availability

The authors declare that they have no competing interests.

Wu Yan conceived and designed the experiments, performed the experiments, analyzed the data, prepared figures and/or tables, authored or reviewed drafts of the article, and approved the final draft.

Li Tan performed the experiments, prepared figures and/or tables, and approved the final draft.

Li Meng-Shan conceived and designed the experiments, prepared figures and/or tables, and approved the final draft.

Sheng Sheng analyzed the data, authored or reviewed drafts of the article, and approved the final draft.

Wang Jun analyzed the data, authored or reviewed drafts of the article, and approved the final draft.

Wu Fu-an analyzed the data, authored or reviewed drafts of the article, and approved the final draft.

The following information was supplied regarding data availability:

The codes, architecture, parameters, dataset, functions, usage and output of the proposed model are available in the Supplemental Files and at GitHub: https://github.com/gnnumsli/-DNA-Sequence-Forecasting-resource.

The data is available at NCBI: NC_001405, NC_010808, NC_048213, NC_010147, NC_011646, NC_028764.

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
