# Peer review of "SaPt-CNN-LSTM-AR-EA: a hybrid ensemble learning framework for time series-based multivariate DNA sequence prediction"

_PeerJ, doi:10.7717/peerj.16192_

## Round 0.1 · original submission · Major Revisions

Please find the reviewer reports attached, both reviewers see the potential in the paper but also raise major concerns regarding its current state. Personally, I agree that the current form is not very approachable and that further clarifications are needed for a better and easier understanding of the approach.

Reviewer 1 ·

Basic reporting

In the paper, biological sequence is represented with time series to form biological time sequence (BTS).

Experimental design

In the DNA sequence experiments of six kinds of viruses, SaPt-CNN-LSTM-AR-EA realized good overall prediction performance, the prediction accuracy and correlation were 1.7073 and 0.9186, respectively.

Validity of the findings

The effectiveness and stability of SaPt-CNN-LSTM-AR-EA were verified through the comparison with other ûve benchmark models. In addition, compared with other benchmark models, SaPt-CNN-LSTM-AR-EA increased the average accuracy by about 30%.

Additional comments

1. The Abstract should be very precise.
2. Motivations of the paper are not clear.
3. Contributions are not specific. The last paragraph of the Introduction should be the structure of the paper. Most importantly, the structure of the Introduction section is very poor.
3. Related schemes are not discussed properly. The following recent papers must be cited to improve related schemes:
a) Probabilistic time series forecasting with deep non-linear state space models
b) Research on face intelligent perception technology integrating deep learning under different illumination intensities
c) Enhanced neural network-based univariate time-series forecasting model for big data
d) Research on trend prediction of component stock in fuzzy time series based on deep forest
e) Enhancing randomness of the ciphertext generated by DNA-based cryptosystem and finite state machine
f) Coupling adversarial learning with selective voting strategy for distribution alignment in partial domain adaptation
g) Ensemble algorithm using transfer learning for sheep breed classification
4. Algorithms of the proposed scheme are not mentioned properly.
5. Why and how DNA computing is used in this paper.
6. The proposed scheme is unstructured. It is hard to identify the novelty of the proposed work.
7. Equations and figures are not represented properly. Use an appropriate software to draw the figures of the result section.
8. Technical discussion on results is not given. Moreover, the results are not convincing.
9. The English language is very poor.
10. The organization of the paper is poor.
11. Important references are missing and all the details of the references are not given.
12. For existing papers, the surname of the authors must be mentioned. Not full author's name.

Reviewer 2 ·

Basic reporting

This paper proposed a hybrid ensemble learning framework for biological time sequence.
1. The contributions and novelty of this study are not clear, compared with the authors' previous study and existing literature. The proposed framework seems to be a simple combination of existing deep learning modules.
2. The mathematical symbols and equations in this manuscript have a terrible format and should be improved.
3. The figures have a low quality and should be improved with a higher resolution.

Experimental design

1. The experiments are not enough. The ablation study is missing in this manuscript.

Validity of the findings

no comment

---

## Round 0.2 · accepted · Accept

As you can see from the reviewer comments, all their concerns have been addressed. The only minor comment is regarding the typos and grammatical mistakes that require still a proof-reading.

Reviewer 1 ·

Basic reporting

In the paper, biological sequences are
represented with time series to obtain biological time sequence (BTS).

Experimental design

Experiments are executed to evaluate the performance of the proposed scheme.

Validity of the findings

Experiment results show the effectiveness of the proposed scheme.

Additional comments

There are some typos and grammatical mistakes. The authors must proofread the entire paper.

Reviewer 2 ·

Basic reporting

no comment

Experimental design

no comment

Validity of the findings

no comment